# Artificial intelligence as a diagnostic aid in cross-sectional radiological imaging of surgical pathology in the abdominopelvic cavity: a systematic review

George E Fowler ,[1] Natalie S Blencowe ,[1] Conor Hardacre,[2] Mark P Callaway,[3] Neil J Smart,[4] Rhiannon Macefield [1]

[1]NIHR Bristol Biomedical Research Centre, Population Health Sciences, Bristol Medical School. University of Bristol, Bristol, UK
[2]Bristol Medical School, University of Bristol, Bristol, UK
[3]Department of Clinical Radiology, University Hospital Bristol and Weston NHS Foundation Trust, Bristol, UK
[4]Exeter Surgical Health Services Research Unit (HeSRU), Royal Devon and Exeter NHS Foundation Trust, Exeter, UK

**Correspondence to**
George E Fowler;
george.fowler@bristol.ac.uk

## ABSTRACT

**Objectives** There is emerging use of artificial intelligence (AI) models to aid diagnostic imaging. This review examined and critically appraised the application of AI models to identify surgical pathology from radiological images of the abdominopelvic cavity, to identify current limitations and inform future research.

**Design** Systematic review.

**Data sources** Systematic database searches (Medline, EMBASE, Cochrane Central Register of Controlled Trials) were performed. Date limitations (January 2012 to July 2021) were applied.

**Eligibility criteria** Primary research studies were considered for eligibility using the PIRT (participants, index test(s), reference standard and target condition) framework. Only publications in the English language were eligible for inclusion in the review.

**Data extraction and synthesis** Study characteristics, descriptions of AI models and outcomes assessing diagnostic performance were extracted by independent reviewers. A narrative synthesis was performed in accordance with the Synthesis Without Meta-analysis guidelines. Risk of bias was assessed (Quality Assessment of Diagnostic Accuracy Studies-2 (QUADAS-2)).

**Results** Fifteen retrospective studies were included. Studies were diverse in surgical specialty, the intention of the AI applications and the models used. AI training and test sets comprised a median of 130 (range: 5–2440) and 37 (range: 10–1045) patients, respectively. Diagnostic performance of models varied (range: 70%–95% sensitivity, 53%–98% specificity). Only four studies compared the AI model with human performance. Reporting of studies was unstandardised and often lacking in detail. Most studies (n=14) were judged as having overall high risk of bias with concerns regarding applicability.

**Conclusions** AI application in this field is diverse. Adherence to reporting guidelines is warranted. With finite healthcare resources, future endeavours may benefit from targeting areas where radiological expertise is in high demand to provide greater efficiency in clinical care. Translation to clinical practice and adoption of a multidisciplinary approach should be of high priority.

**PROSPERO registration number** CRD42021237249.

## STRENGTHS AND LIMITATIONS OF THIS STUDY

⇒ This systematic review examined and critically appraised the application of artificial intelligence models to identify surgical pathology from cross-sectional radiological images, including CT, MR, CT-positron emission tomography and bone scans, and identified current limitations and evidence gaps, and, thereby, focusing future research efforts.

⇒ Robust methodology was undertaken including screening, data extraction and of risk of bias assessment by two independent reviewers.

⇒ Findings may be limited by including English language publications only and incomplete reporting in some of the included studies.

## INTRODUCTION

The widespread adoption of digital healthcare provides vast data to enable the application of artificial intelligence (AI) in pattern recognition.[1] This can alleviate the burden of, or replace, tasks traditionally dependent on clinicians. Examples include the interpretation of medical images for diagnostic, prognostic, surveillance and management decisions, which otherwise rely on a limited number of interpreters and human resources.[2] There has been a surge of research into the use of AI in diagnostic imaging, exploring how it can support clinicians and provide greater efficacy and efficiency in clinical care.[3]

Systematic reviews on the diagnostic accuracy of AI in medical imaging (including respiratory medicine, ophthalmology and breast cancer),[4] gastroenterology,[5] neurosurgery[6] and vascular surgery[7] have demonstrated the diverse application of AI models to detect many pathologies. A variety of imaging modalities have been explored (eg, CT, MR and positron emission tomography (PET)). AI models can demonstrate diagnostic performances equivalent to that of experts[8] and with greater efficiency, for example, in

the time taken to diagnose childhood cataracts (5.7 min quicker than senior consultants).[9] However, while AI technologies offer to markedly reduce the clinical workload, 'black box' techniques (AI algorithms) can be difficult or impossible to interpret, which can be a barrier to adopting these techniques into clinical practice. Furthermore, many AI studies are proof-of-concept[10] and poorly reported,[11 12] including limited details on participants, making it difficult to replicate or interpret the study findings.[12]

A systematic review of the diagnostic accuracy of AI models in cross-sectional radiological imaging of the abdominopelvic cavity is lacking.[13] Synthesis of the current AI research in this area could benefit several different surgical specialities which image this region, such as endocrine surgery, gastrointestinal surgery, obstetrics and gynaecology, urology and vascular surgery to guide their clinical decision-making. This study aimed to conduct a systematic review to examine and critically appraise the application of AI models to identify surgical pathology from cross-sectional radiological images, including CT, MR, CT-PET and bone scans, of the abdominopelvic cavity, to identify current limitations and inform future research efforts.

## METHODS

### Protocol and registration

This systematic review was registered with the International Prospective Register of Systematic Reviews. A study protocol has previously been published.[13] It is reported in accordance with the Preferred Reporting Items for Systematic Reviews and Meta-analyses of Diagnostic Test Accuracy Studies.[14]

### Information sources

Electronic searches of OVID SP versions of Medline, EMBASE and the Cochrane Central Register of Controlled Trials databases were conducted to identify all potentially relevant studies. Date limitations between 1 January 2012 and 31 July 2021 were applied, to account for advancements in machine learning performance and the development of deep learning approaches since 2012, in line with existing reviews.[15] Reference lists of included articles were screened to identify further relevant studies.

### Search strategy and study selection

A comprehensive search syntax was developed with adaptation from three existing search strategies[6 7 16] and guidance from an information specialist using text words and medical subject headings related to three domains: 'artificial intelligence', 'diagnostic imaging' and the 'abdominopelvic cavity' (online supplemental appendix S1). Database search results were imported into reference management software (EndNote V.X9, Clarivate Analytics, USA) and duplicates were removed.

Assessment of study eligibility was performed in two stages. First, titles and abstracts were screened for inclusion by two independent reviewers (GEF and CH). Any conflicts were resolved through discussion, referring to the wider study team if required. Final eligibility was assessed by a full-text review of potentially eligible studies by the same process. Management of the screening process was aided by Rayyan software (Rayyan Systems, Cambridge, Massachusetts).[17]

### Eligibility criteria

Primary research studies were considered for eligibility using the PIRT (participants, index test(s), reference standard and target condition) framework.[18] Participants were adults with pathology within the abdominopelvic cavity diagnosed using the following radiological modalities: CT, MR, CT-PET or bone scans. Diagnostic endoscopy was excluded as existing reviews have explored the performance of AI models in this area.[19 20] The index test was studies considering AI models as an intervention with the aim to provide a diagnosis. The reference standard was 'standard practice' to allow for variation across the included studies. The target condition was abdominopelvic cavity pathology which has had, or may warrant, an invasive procedure[21] for therapeutic intent.

Excluded were secondary research studies (eg, systematic reviews), case reports and case series, absence of full text (eg, conference abstracts), animal studies and non-English articles.

### Data extraction and management

Data extraction from the included articles was independently performed by two reviewers (GEF and CH). Data management software (REDCap V.9.5.23, Vanderbilt University, USA)[22] and a predesigned standardised form were used. Data were extracted under the following three subheadings:

#### Study characteristics

Extracted data included the name of the first author, their affiliated country, composition of the study team (eg, software engineers, radiologists and surgeons who routinely operate on surgical pathology within the abdominopelvic cavity (eg, gastrointestinal surgeons, urologists and gynaecologists)), year of publication, study aim and design (ie, 'prospective' or 'retrospective'), surgical subspeciality and pathology studied (ie, benign, malignant tumours, multiple or other). Information on the reporting of ethics and/or regulatory approval (eg, Medicines and Healthcare products Regulatory Agency), patient and publication involvement and authors' mention of using a reporting guideline (eg, Standards for Reporting of Diagnostic Accuracy Studies) was recorded.

#### Training data

Extracted data on the input features (data used to develop the AI model) included the modality of cross-sectional radiological imaging (CT, MR, CT-PET and bone scans), the AI model used, the reference standard and the reporting and size of the training and test sets.

Information on whether the training data came from the studies dataet or publicly available datasets was recorded.

## Outcomes

The performance of the AI models and human comparator (where applicable) was extracted. Diagnostic measures of accuracy included reported sensitivity, specificity, positive predictive values and the area under the receiver operating characteristic curve (AUC). The interpretation time (seconds) for studies comparing the performance of the AI model with a human comparator (where reported) was extracted.

## Risk of bias and applicability

Risk of bias was assessed independently by two reviewers (GEF and RM) using the **Q**uality **A**ssessment of **D**iagnostic **A**ccuracy **S**tudies-2 (QUADAS-2) tool.[23] A version of the QUADAS-2 tool for AI studies[24] was still in development and not available at the time of conducting the current review. The generic QUADAS-2 tool with the pre-existing modified signalling questions was used to assess four domains including patient selection, index test, reference standard and flow and timing.[23] An overall judgement of 'at risk of bias' or 'concerns regarding applicability' was assigned if more one or more domains were judged as 'high' or 'unclear'.[23] Judgements of applicability assessed whether the study matched the review question.[23]

## Data synthesis

A narrative synthesis was conducted according to the Synthesis Without Meta-analysis guidelines.[25] The synthesis was planned to focus on the primary outcome with studies grouped by the modality of radiological imaging, surgical subspeciality and pathology studied, as outlined in the protocol.[13] A broader approach was, however, adopted due to the small number of included studies and their heterogeneity. A meta-analysis was not performed due to the broad nature of the included studies.

## Patient and public involvement

As part of the wider programme of work (Bristol Biomedical Research Centre, National Institute for Health Research Bristol BRC), patients and the public were consulted on their views of AI being used to guide doctors to make decisions about treatment. Overall, it was perceived positively, and they were supportive of its adoption in healthcare.

## RESULTS

Database searching identified 628 records, with a further five studies identified through reference lists of included articles. After the removal of duplicates, 580 were screened and, 52 full-text articles were assessed for eligibility. Fifteen studies were finally included (figure 1).[26–40]

## Study characteristics

Characteristics of the included studies and details of the AI models are summarised (tables 1 and 2). All were retrospective studies conducted in six different countries: Japan (n=4), USA (n=4), China (n=3), India (n=2), Turkey (n=1) and South Korea (n=1). Studies were all proof-of-concept (ie, not applied in a clinical setting), from four surgical specialities: urology (n=6), gastrointestinal surgery (n=6), endocrine surgery (n=1) and gynaecology (n=1). One study was not specific to a

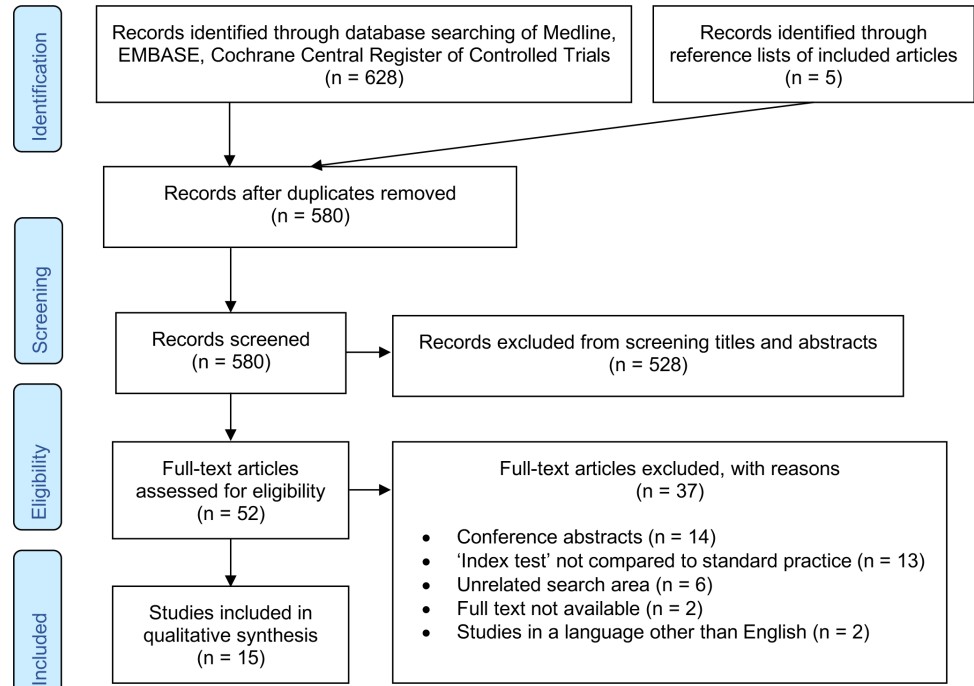

**Figure 1** PRISMA flow diagram. PRISMA, Preferred Reporting Items for Systematic Reviews and Meta-Analysis.

**Table 1** Characteristics of the included studies

| Characteristics | Number of studies n=15 |
|---|---|
| Country of origin | |
| Japan | 4 |
| USA | 4 |
| China | 3 |
| India | 2 |
| Turkey | 1 |
| South Korea | 1 |
| Year of publication | |
| 2020 | 5 |
| 2019 | 5 |
| 2018 | 2 |
| 2017 | 1 |
| 2016 | 1 |
| 2013 | 1 |
| Surgical subspecialty | |
| Urology | 6 |
| Gastrointestinal | 6 |
| Endocrine | 1 |
| Gynaecology | 1 |
| Not reported | 1 |
| Pathology studied | |
| Malignant tumours | 11 |
| Multiple pathologies | 2 |
| Other | 2 |
| Modality of radiological imaging | |
| CT | 8 |
| CT-PET | 2 |
| MR | 2 |
| PET | 1 |
| MR and CT-PET | 1 |
| Bone scans | 1 |

PET, positron emission tomography.

single specialty and involved whole-body CT-PET across four anatomical regions (head-and-neck, chest, abdomen and pelvis).[29] Most studies focused on malignant tumours (n=11) using CT as the imaging modality (n=8). Most studies (n=14) included an ethical approval statement. No studies, however, mentioned patient and public involvement or the use of a reporting guideline.

The composition and expertise within each study team varied. Three studies (n=3) comprised teams including software engineers, radiologists and surgeons. A small number of studies were comprised of only radiologists (n=2) or software engineers alone (n=1). Four study teams were comprised software engineers and radiologists together. Of the remaining five studies, three had no apparent radiological team and two were comprised of radiologists with either a team of software engineers, surgeons or physicians.

### Training data

AI training and test sets within the included studies comprised a median of 130 (range: 5–2440) and 37 (range: 10–1045) patients, respectively (table 2). This information, however, was not always available (n=6 studies). All training data came from the studies collected data and not from pre-existing readily available data sets. There was variability in the reference standard, including the number of clinicians (range: 1–3) for either a radiological (n=9) or histological (n=4) diagnosis or both (n=1). Only one study had an unclear reference standard (table 2).

### Outcomes

The intent of the AI applications in the included studies varied, with the majority focusing on diagnosing advanced or recurrent cancer (n=6 studies) and four studies classifying the pathology (ie, normal or abnormal, or benign or malignant) (online supplemental table 1). Diagnostic performance of the AI models ranged between 70.0% and 95.0% sensitivity, and 52.9% and 98.0% specificity (online supplemental table 1). The reporting of the diagnostic measures of accuracy was unstandardised, for example, there were different output measures across the studies, and three studies did not report all of their outcome measures (online supplemental table 1).

Following model development with a training and tuning set, four studies had an external validation test set and compared the performance of the AI model with a human comparator (a radiologist) (online supplemental table 1).[32 33 39 40] The diagnostic performance of the radiologists ranged between 57.4% and 62.8% sensitivity and 0.89 and 0.97 AUC (online supplemental table 1). In these studies, there was variability in both the number of patients (range 50–414; from one to six different centres) and number of radiologists (range 2–4) and their years of experience. These studies reported the diagnostic performance of the AI model as either superior (n=2; in rectal and advanced gastrointestinal cancer)[39 40] or comparable (n=2; in metastatic gastrointestinal pathology and a gynaecology study distinguishing between malignant tumours and benign pathology) to radiologists.[32 33] Two of these studies included a comparison of the interpretation time of the AI model with the radiologists.[32 40] AI models outperformed the radiologists in both studies (1–2 s vs 200 s per case, $p < 0.05$[40] and 20 s vs 600 s for an average of 100 MRIs[32]).

### Risk of bias and applicability

With the exception of one study,[40] all the included studies had an overall judgement of 'at risk of bias' and 'concerns regarding applicability' (table 3). This was predominantly due to comparisons based on either a single clinician's assessment (n=10) or an unclear number of clinicians for the assessment (n=5), from either a small (eg, 10 patients

**Table 2** Study details and AI models

| First author (Reference) | Country of origin | Surgical specialty | Pathology studied | Modality of radiological imaging | AI models | Reference standard | Size of training set (patients) | Size of test set (patients) |
|---|---|---|---|---|---|---|---|---|
| Acar et al[26] | Turkey | Urology | Malignancy | CT-PET | KNN | Two nuclear medicine physicians | 90% (total 75 patients) | 10% (total 75 patients) |
| Coy et al[27] | USA | Urology | Malignancy | CT | CNN | Pathologist (number not specified) | 90% (total 179 patients) | 10% (total 179 patients) |
| Han et al[28] | South Korea | Urology | Malignancy | CT | CNN | Pathologist (number not specified) | 135 | 34 |
| Koizumi et al[30] | Japan | Urology | Malignancy | Bone scans | ANN | Two radiology nuclear physicians | Not reported | 226 |
| Lee et al[31] | USA | Urology | Malignancy | PET | CNN | One imaging physician | Not reported | Not reported |
| Oberai et al[35] | USA | Urology | Malignancy | CT | CNN | One genitourinary pathologist | 120 | 23 |
| Lu et al[32] | China | Gastrointestinal | Malignancy | MR | CNN | Radiologists (number not specified) | Not reported | Not reported (+ 414 validation test set) |
| Nayak et al[34] | India | Gastrointestinal | Multiple (cirrhosis, hepatocellular carcinoma) | CT | SVM | One radiologist | 40 | Not reported |
| Sethi et al[37] | India | Gastrointestinal | Multiple (normal, cystic, calculus or tumour tissues) | CT | ANN, SVM | Unclear | Not reported | Not reported |
| Yasaka et al[38] | Japan | Gastrointestinal | Malignancy | CT | CNN | Radiologists (number not specified) | 460 | 100 |
| Yuan et al[39] | China | Gastrointestinal | Malignancy | CT | SVM, CNN | One pathologist | 130 | 40 |
| Zhao et al[40] | China | Gastrointestinal | Malignancy | MR | CNN | Three radiologists | 293 | 81 |
| Saiprasad et al[36] | USA | Endocrine surgery | Other (normal vs abnormal) | CT | RFC | Two radiologists | 5 | 10 |
| Nakagawa et al[33] | Japan | Gynaecology | Malignancy | MR and CT-PET | Logistic regression (univariate and multivariate) | Two radiologists and pathologists (number not specified) | 66 | Unclear |
| Kawauchi et al[29] | Japan | Uncharacterised | Other (benign/malignant/equivocal) | CT-PET | CNN | One nuclear medicine physician | 2440 | 1045 |

AI, artificial intelligence; ANN, artificial neural network; CNN, convolutional neural network; KNN, k-nearest neighbours algorithm; PET, positron emission tomography; RFC, random forest classification; SVM, support vector machine.

**Table 3**  Study quality assessment (QUADAS-2 tool)

| First author (reference) | Risk of bias | | | | Applicability concerns | | |
|---|---|---|---|---|---|---|---|
| | Patient selection | Index test | Reference standard | Flow and timing | Patient selection | Index test | Reference standard |
| Acar et al[26] | Low | Low | High | Low | Low | Low | High |
| Coy et al[27] | Low | Low | High | Low | Low | Low | High |
| Han et al[28] | Unclear | Low | High | Low | Low | Low | High |
| Kawauchi et al[29] | Low | Low | High | Low | Low | Low | High |
| Lu et al[32] | Unclear | Low | High | Low | Low | Low | High |
| Nakagawa et al[33] | Low | Low | High | Low | Low | Low | High |
| Koizumi et al[30] | Low | Low | High | Low | Low | Low | High |
| Lee et al[31] | Low | Low | High | Low | Low | Low | High |
| Nayak et al[34] | Unclear | Low | Unclear | Low | Low | Low | High |
| Oberai et al[35] | Low | Low | High | Low | Low | Low | High |
| Yasaka et al[38] | Unclear | Low | High | Low | Low | Low | High |
| Saiprasad et al[36] | Unclear | Low | High | Low | Low | Low | High |
| Sethi et al[37] | High | Low | Unclear | Low | Low | Low | Unclear |
| Yuan et al[39] | Low | Low | High | Low | Low | Low | High |
| Zhao et al[40] | Low | Low | Low | Low | Low | Low | Low |

QUADAS-2, **Q**uality **A**ssessment of **D**iagnostic **A**ccuracy **S**tudies-2.

in one study) or unclear size of test set, and most models were developed only from internal validations (n=11) (table 2 and online supplemental table 1).

## DISCUSSION

The major finding of this review was the heterogeneity in the AI applications across the included studies regardless of the surgical specialty or pathology. Early phase studies of AI innovation, particularly focusing on advanced or recurrent malignancy, were identified with promising diagnostic accuracies to support clinical decision-making. Future AI research could benefit from targeting areas where radiological expertise is in high demand or the data are complex to interpret; for example, adrenal incidentalomas[41] and images from virtual colonoscopy.[2] Attention should also be directed to the governance of AI, particularly on where the responsibility lies if the AI model misses a lesion.

In this review, several reporting issues were identified, including for the reference standard and training data. Poor adherence to reporting guidelines is a common finding in the existing literature for diagnostic accuracy studies assessing AI interventions.[4 11] The Standards for Reporting of Diagnostic Accuracy-AI (STARD-AI) Steering Group are developing an AI-specific extension to the STARD statement, which aims to improve reporting of AI diagnostic accuracy studies.[42] This steering group highlighted three pitfalls, which are also reflected in this review: (1) unclear methodological interpretation (eg, methods of validation and comparison to human performance), (2) unstandardised nomenclature (eg, varying definition

of the term 'validation') and (3) heterogeneity of the outcome measures (eg, sensitivity, specificity, predictive values and AUC).[43] Endeavours to address this include the development of specific reporting guidelines for authors of AI studies, including protocols (SPIRIT-AI),[44] reports (CONSORT-AI)[45] and proposals (**MIN**imum **I**nformation for **M**edical **AI R**eporting (MINIMAR)).[46] These efforts should improve both the reporting quality and make it easier to interpret and compare AI studies.

A minority of the included studies compared the diagnostic performance of the AI model with a clinician's diagnosis. These studies reported a faster and superior or equivalent diagnostic performance with the AI model. A recent review found only 51 studies worldwide reporting the implementation and evaluation of AI applications in clinical practice.[47] While many AI studies are currently retrospective and proof-of-concept,[10 11 47] which may be appropriate for early phase surgical research, future efforts should evaluate the role of AI in a clinical setting. This should adopt a multidisciplinary team of all relevant stakeholders (eg, software engineers, radiologists and surgeons) to ensure that the diverse and relevant skill sets can work together to produce both high-quality and clinically relevant AI research.

This review included a robust methodology, comprehensive search strategy and a multidisciplinary team. Some limitations, however, are acknowledged. Despite having a broad search strategy, relevant studies may have been missed by excluding articles that were not published in the English language. It did not encompass all diagnostic applications of AI in this region, such as diagnostic

imaging for prognostic, surveillance and management decisions meaning findings are not generalisable to these wider contexts. However, recommendations for prioritising future endeavours on clinical need, adhering to reporting guidelines and standardised and transparent reporting can be considered appropriate for all studies assessing AI interventions in healthcare.

This review identified a diverse application of AI innovation in this field. Most studies were proof-of-concept and more 'comparator' studies in the clinical setting are needed. Future AI research could build on existing studies with translation to clinical practice, adopting a multidisciplinary approach, including patient and public involvement, which was lacking in the studies of this review. This could target areas of clinical need. Adherence to existing and developing guidelines for reporting AI studies, such as SPIRIT-AI,[44] CONSORT-AI,[45] STARD-AI[43] and DECIDE-AI,[48] is warranted.

**Acknowledgements** The authors would like to thank Ms Catherine Borwick, Information Specialist, University of Bristol, for her input and expertise to develop the search strategy. The authors are grateful to Dr Christin Hoffmann (University of Bristol) for leading patient and public feedback sessions as part of the wider programme of this research and Dr Xiaoxuan Liu (University of Birmingham and University Hospitals Birmingham NHS Foundation Trust) for providing expert comments on the manuscript.

**Contributors** GEF and NSB conceived the idea for this systematic review. All authors (GEF, NSB, CH, MPC, NS and RM) contributed to the design of the study. GEF and CH performed screening and data extraction. GEF and RCM conducted the QUADAS-2 assessments. All authors contributed towards the data analysis and interpretation of the data. GEF drafted the manuscript (guarantor of review) and all authors were involved in critiquing the manuscript. All authors approved the final manuscript before submission.

**Funding** NSB is funded by an MRC Clinical Scientist Award (grant number: MR/S001751/1). This study was supported by the National Institute for Health and Care Research Bristol Biomedical Research Centre. The views expressed are those of the authors and not necessarily those of the NIHR or the Department of Health and Social Care.

**Competing interests** None declared.

**Patient and public involvement** Patients and/or the public were not involved in the design, or conduct, or reporting, or dissemination plans of this research.

**Patient consent for publication** Not applicable.

**Ethics approval** Ethical approval was not required as no primary data were collected.

**Provenance and peer review** Not commissioned; externally peer reviewed.

**Data availability statement** All data relevant to the study are included in the article or uploaded as supplementary information.

**ORCID iDs**
George E Fowler http://orcid.org/0000-0002-4133-802X
Natalie S Blencowe http://orcid.org/0000-0002-6111-2175
Rhiannon Macefield http://orcid.org/0000-0002-6606-5427

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
