## [Reviewer comments · BMJ Open]

ARTICLE DETAILS

TITLE (PROVISIONAL)	Artificial intelligence as a diagnostic aid in cross-sectional radiological imaging of surgical pathology in the abdominopelvic cavity: a systematic review
AUTHORS	Fowler, George; Blencowe, Natalie; Hardacre, Conor; Callaway, Mark; Smart, Neil; Macefield, Rhiannon

VERSION 1 – REVIEW

REVIEWER	Yeo, Anthony Western Sydney University
REVIEW RETURNED	10-Jul-2022

GENERAL COMMENTS	For supplementary Table 1, the abbreviation for ANN is not listed. The primary issue of this paper is that it only looked at 15 studies (even though the authors started with a large number). This is thus a small study. Other than this, the authors did systematically appraise the literature in the required manner and it was sufficiently scholarly. Given this small number of studies, I recommend that it be published as a short article as opposed to a full paper, if this is permitted.
--

REVIEWER	Oo, Aung Myint Tan Tock Seng Hospital, General Surgery
REVIEW RETURNED	18-Jul-2022

GENERAL COMMENTS	Dear authors, Thanks for the interesting review. Few comments page 19 line 17 should be "or" instead of "of" title is cross sectional imaging. however some of the studies included didn't mention that the cross sectional images were used in the data set study 25 and 34 were about bone metastasis and 34 used bone scintigraphy (suggest to include in table 1 description) suggest to change the title and content to be more general to reflect the studies included in the review. All the best. Best regards.
--

REVIEWER	Leung, Siu-wai University of Edinburgh Western General Hospital, Edinburgh Bayes Centre for AI Research in Shenzhen
REVIEW RETURNED	20-Jul-2022

GENERAL COMMENTS	The manuscript reads well as a systematic review. The following are suggestions for a minor revision:  - This systematic review followed PRISMA-DTA and SWiM guidelines. Please provide the SWiM checklist (if available). - The PRISMA-DTA for Abstracts item “strengths and limitations of the evidence” was not addressed. - Evidence for the optimality of the search strategy should be shown. How does this search strategy compare to other AI systematic reviews? - The search results from individual databases should be reported in PRISMA flow diagram (Figure 1) or in main text so that the reproducibility of the search can be evaluated. - In Figure 1, the 528 records excluded from the screening procedure should be categorized in terms of reasons for exclusion. - In Figure 1, the expressions such as “Conference abstract = 14” can be improved to read “Conference abstracts (n = 14)”. - Did the authors of eligible studies collect their imaging data? Or they could use readily available datasets? Please make it explicit in the eligibility criteria. In case existing datasets were eligible, did this systematic review conduct any checking of veracity/reliability of the data in the included studies? - It would be tricky to compare the performances between humans and AI. Were there any specific misleading comparisons (or misconceptions) found from the eligible studies? - Were there any prospective studies on comparing human and AI performances in DTA? - What would be evidence level and evidence strength of the eligible studies (as seen from critical appraisal) to support the AI performance? - Please also elaborate the best practices (including reporting guidelines) that can be drawn from the eligible studies and are recommended by this systematic review? - As seen from the contributorship statement, some authors might not fulfil the ICMJE authorship criteria. Please check and verify.
---

VERSION 1 – AUTHOR RESPONSE

Reviewer #1:

1. For supplementary Table 1, the abbreviation for ANN is not listed.

Response: Thank you for highlighting this. We have now included the abbreviation in the table footnote.

2. Primary issue of this paper is that it only looked at 15 studies (even though the authors started with a large number). This is thus a small study. Other than this, the authors did systematically appraise the literature in the required manner and it was sufficiently scholarly. Given this small number of studies, I recommend that it be published as a short article as opposed to a full paper, if this is permitted.

Response: We acknowledge that 15 studies may be considered to be a relatively small number of included articles. This number of studies, however, is similar to other published systematic reviews that focus on a specific field of AI innovation, for example, AI for polyp detection during colonoscopy (N = 5) [1], AI for detection of anterior cruciate ligament and meniscus tears (N = 11) [2] and the

neurosurgical applications of machine learning (N = 23) [3]. We wish to publish the study as a systematic review so that the full details are available for the readers.

Reviewer #2:

1. page 19 line 17 should be "or" instead of "of"

Response: We apologise for this typo and have amended this mistake accordingly.

2. title is cross sectional imaging. however some of the studies included didn't mention that the cross sectional images were used in the data set. study 25 and 34 were about bone metastasis and 34 used bone scintigraphy (suggest to include in table 1 description) suggest to change the title and content to be more general to reflect the studies included in the review.

Response: We thank the reviewer for raising this observation. As suggested, we have now included the descriptions in Table 1 to describe the studies more clearly to the reader. We agree that it would be helpful to be more explicit that the cross-sectional imaging included bone scans in this review. We have therefore amended and elaborated on this in the manuscript in several places, including the abstract, aims and eligibility criteria/study selection.

Reviewer #3:

The manuscript reads well as a systematic review. The following are suggestions for a minor revision:

1. This systematic review followed PRISMA-DTA and SWiM guidelines. Please provide the SWiM checklist (if available).

Response: Thank you for this suggestion. We have now included the SWiM checklist.

2. The PRISMA-DTA for Abstracts item "strengths and limitations of the evidence" was not addressed.

Response: Thank you for highlight this and we have now included a key strength (data extraction was performed by independent reviewers) and a key limitation (English language publications only) in the abstract. Unfortunately, we are restricted by word count to explicitly include any more of the strengths and limitations in the abstract. We are however also including strengths and limitations as specific bullet points as per the journals format, so these will be published alongside our abstract.

3. Evidence for the optimality of the search strategy should be shown. How does this search strategy compare to other AI systematic reviews?

Response: Thank you for this suggestion. The search strategy in this review was developed with adaptation from three existing search strategies [3,5,6] and guidance from an information specialist. This has been made clearer in the methods section of the manuscript ('search strategy and study selection').

4. The search results from individual databases should be reported in PRISMA flow diagram (Figure 1) or in main text so that the reproducibility of the search can be evaluated.

Response: We thank the reviewer for this suggestion. For clarity, we have now included the specific databases searched in the PRISMA flow diagram. We have chosen to report combined number of records identified. The PRISMA guidelines [7] suggest the number of records could be reported from the individual databases, however, this detail is not essential to report. We have included the search strategy for the individual databases as supplementary material making it readily available for anyone wishing to reproduce the search.

5. In Figure 1, the 528 records excluded from the screening procedure should be categorized in terms of reasons for exclusion.

Response: Thank you for this comment. We have reported the number of records excluded from screening titles and abstracts, with reasons for exclusion provided at full assessment in line with the PRISMA guidelines.

6. In Figure 1, the expressions such as “Conference abstract = 14” can be improved to read “Conference abstracts (n = 14)”.

Response: Thank you for this suggestion which we have adopted.

7. Did the authors of eligible studies collect their imaging data? Or they could use readily available datasets? Please make it explicit in the eligibility criteria. In case existing datasets were eligible, did this systematic review conduct any checking of veracity/reliability of the data in the included studies?

Response: Primary collection of imaging data in the individual studies was not a criteria for eligibility and included studies could use existing datasets. However, all included studies collected imaging data as part of the study and none used existing datasets. No checking of veracity/reliability was therefore required.

We have chosen to report this in the methods section under ‘data extraction and management’ rather than in the eligibility criteria as it reflects when this information was recorded, as it did not influence the screening of articles when assessing eligibility.

8. It would be tricky to compare the performances between humans and AI. Were there any specific misleading comparisons (or misconceptions) found from the eligible studies?

Response: Four of the 15 included studies compared the performances between humans and AI. No misleading comparisons (or misconceptions) were found in these studies.

9. Were there any prospective studies on comparing human and AI performances in DTA?

Response: The fifteen included studies were all retrospective and no prospective studies comparing human and AI performances in DTA were included. We have included the description of the study type in the results section (‘study characteristics’).

10. What would be evidence level and evidence strength of the eligible studies (as seen from critical appraisal) to support the AI performance?

Response: We only looked at risk of bias (QUADAS-2) to assess the level and strength of evidence within studies. The review focused on the application of AI models and not their performance as part of a meta-analysis.

11. Please also elaborate the best practices (including reporting guidelines) that can be drawn from the eligible studies and are recommended by this systematic review?

Response: Guidelines for best practice have recently become available, for example, SPIRIT-AI and CONSORT-AI, and there are more recommendations in development, such as the Standards for Reporting of Diagnostic Accuracy-AI (STARD-AI). We are recommending future studies follow these guidelines for best practices. We have included details and references in our discussion and we have now also amended our conclusion to explicitly point the reader to these available guidelines.

12. As seen from the contributorship statement, some authors might not fulfil the ICMJE authorship criteria. Please check and verify.

Response: We can confirm that all authors meet the ICMJE authorship criteria. We have added more detail to our contributorship statement to describe the authors' contributions and apologise that this was not previously clear.

We hope that our responses to these helpful comments and the revisions to the manuscript are satisfactory to address the reviewers comments. We look forward to hearing from you.

Yours sincerely,

Dr George E Fowler

NIHR Bristol Biomedical Research Centre, Population Health Sciences, Bristol Medical School.
University of Bristol, UK.

Email: George.fowler@bristol.ac.uk

References

- 1 Barua I, Vinsard DG, Jodal HC, et al. Artificial intelligence for polyp detection during colonoscopy: a systematic review and meta-analysis. *Endoscopy* 2021;53:277–84. doi:10.1055/a-1201-7165
- 2 Kunze KN, Rossi DM, White GM, et al. Diagnostic Performance of Artificial Intelligence for Detection of Anterior Cruciate Ligament and Meniscus Tears: A Systematic Review. *Arthroscopy: The Journal of Arthroscopic & Related Surgery* 2021;37:771–81. doi:10.1016/j.arthro.2020.09.012
- 3 Senders JT, Arnaout O, Karhade A v., et al. Natural and artificial intelligence in neurosurgery: A systematic review. *Clin Neurosurg* 2018;83:181–92. doi:10.1093/neuros/nyx384
- 4 Cohen JF, Deeks JJ, Hooft L, et al. Preferred reporting items for journal and conference abstracts of systematic reviews and meta-analyses of diagnostic test accuracy studies (PRISMA-DTA for Abstracts): checklist, explanation, and elaboration. *BMJ* 2021;;n265. doi:10.1136/bmj.n265
- 5 Yang Y, Jin G, Pang Y, et al. The diagnostic accuracy of artificial intelligence in thoracic diseases: A protocol for systematic review and meta-analysis. *Medicine (United States)* 2020;99:1–5. doi:10.1097/MD.00000000000019114
- 6 Raffort J, Adam C, Carrier M, et al. Artificial intelligence in abdominal aortic aneurysm. *J Vasc Surg* 2020;72:321-333.e1. doi:10.1016/j.jvs.2019.12.026
- 7 McInnes MDF, Moher D, Thombs BD, et al. Preferred Reporting Items for a Systematic Review and Meta-analysis of Diagnostic Test Accuracy Studies The PRISMA-DTA Statement. *JAMA - Journal of the American Medical Association* 2018;319:388–96. doi:10.1001/jama.2017.19163

VERSION 2 – REVIEW

REVIEWER	Leung, Siu-wai University of Edinburgh Western General Hospital, Edinburgh Bayes Centre for AI Research in Shenzhen
REVIEW RETURNED	11-Oct-2022

GENERAL COMMENTS	Thanks for providing additional information and explanation to address my concerns. There's only a few minor comments that should be clarified: (1) Figure 1: The diagram still only reports a combined number of articles from multiple databases. Without the number of articles searched from each database, it is difficult to check the reproducibility of the search. (2) The authors only mentioned that the English language publications as the key limitation. However, it seems that the authors' explanation for the incomplete information from the study would be the limitations. (3) The reporting guidelines such as SPIRIT-AI, CONSORT-AI, and STARD-AI were developed to address the pre-existing problems/issues due to inadequate reporting. Such reporting requirements should not be presented in a way to seem that the requirements are new (actually not new) to researchers in the field. (4) Thanks for the clarification about the sources of the search strategy. It would be nice to mention about how to combine the multiple sources and validate them before use.
--

VERSION 2 – AUTHOR RESPONSE

Reviewer #3:

(1) Figure 1: The diagram still only reports a combined number of articles from multiple databases. Without the number of articles searched from each database, it is difficult to check the reproducibility of the search.

We thank the reviewer for wanting to ensure our search is reproducible. We have followed the PRISMA guidelines which do not mandate the number of records to be reported from the individual databases. However, we have endeavoured to include this information. Unfortunately the breakdown by database for the original search is not available (this part of the work was performed by our Information Specialist, who provided us with the combined result and the breakdown from the original search is not available retrospectively). We repeated the search using the retrospective

dates with the following findings: Medline (n = 85), Embase (n = 555) and Cochrane Central Register of Controlled Trials (n = 8). However, the total is twenty articles more than the original total (n = 628), likely because of the inclusion of additional articles to the respectful databases after the original search date. This discrepancy prevents us from including the breakdown in the manuscript because it is not reflective of the original search. We have included the search strategy from the three databases as supplementary material to ensure the search strategy is reproducible for anyone wanting to use it in future and hope that this is satisfactory for the reviewer.

(2) The authors only mentioned that the English language publications as the key limitation. However, it seems that the authors' explanation for the incomplete information from the study would be the limitations.

Thank you for highlighting this limitation and we agree this is important to emphasise. We have now included this in the main: 'Strengths and limitations of this study' section after the abstract (p. 5). The text now reads as follows: "*Findings may be limited by including English language publications only and incomplete reporting in some of the included studies.*" (p. 5)

(3) The reporting guidelines such as SPIRIT-AI, CONSORT-AI, and STARD-AI were developed to address the pre-existing problems/issues due to inadequate reporting. Such reporting requirements should not be presented in a way to seem that the requirements are new (actually not new) to researchers in the field.

Thank you for this comment. We have edited the wording in our conclusion so that it is clearer to the reader that the guidelines are not all new, as indeed, some are existing, while others are being developed. The text now reads as follows "*Adherence to existing and developing guidelines for reporting AI studies, such as SPIRIT-AI, CONSORT-AI, STARD-AI and DECIDE-AI are warranted.*" (p.23)

(4) Thanks for the clarification about the sources of the search strategy. It would be nice to mention about how to combine the multiple sources and validate them before use.

Our search strategy was developed based on three existing search strategies and guidance from an Information Specialist, with expertise in conducting systematic reviews. We acknowledge that we did not validate the search strategy, which is recommended by Cochrane when developing search strategies for RCTs. We are grateful for your comment and will endeavour to do this in future work.

Yours sincerely,

Dr George E Fowler

NIHR Bristol Biomedical Research Centre, Population Health Sciences, Bristol Medical School.
University of Bristol, UK.

Email: George.fowler@bristol.ac.uk